# THE PREIMAGE OF RECTIFIER NETWORK ACTIVITIES

**Stefan Carlsson, Hossein Azizpour and Ali Razavian**
School of Computer Science and Communication
KTH
Stockholm, Sweden
email stefanc@kth.se, azizpour@kth.se, razavian@kth.se

## ABSTRACT

The preimage of the activities of all the nodes at a certain level of a deep network is the set of inputs that result in the same node activity. For fully connected multi layer rectifier networks we demonstrate how to compute the preimages of activities at arbitrary levels from knowledge of the parameters in a deep rectifying network by disregarding the effects of max-pooling. If the preimage set of a certain activity in the network contains elements from more than one class it means that these classes are irreversibly mixed. This implies that preimage sets which are piecewise linear manifolds are building blocks for describing the input manifolds specific classes, i.e. all preimages should ideally be from the same class. We believe that the knowledge of how to compute preimages will be valuable in understanding the efficiency displayed by deep learning networks and could potentially be used in designing more efficient training algorithms

## 1 INTRODUCTION

The activity of the nodes at each level of a deep network contains all the information that will be used for classification. Ideally, if the activities are generated by two inputs from the same class, they should be similar and if the classes are distinct the activities should be distinct. The map from the input to any layer of a deep network can however easily be shown to be many to one. This means that classes can potentially get mixed at any level of the network by mapping to the same activity. This mixing cannot be undone at later stages and must therefore be avoided. Given a certain activity it is therefore essential to know the set of inputs to the network that result in this activity. This set should obviously not contain exemplars from more than one class. This means that they are potential building blocks for designing efficient classifiers. In this paper we will demonstrate that the set of inputs resulting in a specific activity at any level of a deep rectifier network can be completely characterised and we will give a procedure for computing them. For a specific activity at any level they are known as the *preimage* of the function mapping the input to the activity. In this procedure we disregard the effects of pooling the outputs of node activities. This can be seen as complementary to the work in Mahendran & Vedaldi (2015; 2016) where specific preimages are computed numerically by a regularised optimisation procedure that tries to map the image to the natural image manifold.

For multi layer rectifier networks where each layer consists of linear mappings followed by a rectifying linear unit (ReLU), the set of possible functions that map inputs to node activities can be shown to be piecewise linear Glorot et al. (2011); Montufar et al. (2014). We will demonstrate that for a specific activity at any level, the equivalence class of inputs that can generate this activity consists of piecewise linear manifolds in the input space. For efficient classification by the network, these manifolds must only contain a single class. They therefore constitute building blocks for efficient approximation of the distribution of classes in the input space.

Multi layer networks with rectifier linear units (ReLU) as non-linear elements starts with the $n$-dimensional input vector $x$ and produces successive outputs of the form:

$$y = [\sum_{i=1}^{n} a_i x_i + b]_+ \tag{1}$$

Where $[x]_+$ denotes the ReLU function $max(0, x)$. By augmenting the input vector with a one $x^T = (x_1 \ldots x_n, 1)$ we can absorb the bias $b$ into the vector as a weight $a_{n+1}$ and write

$$y \;=\; [\sum_{i=1}^{n+1} a_i x_i]_+ = [w^T x]_+ \tag{2}$$

where we have collected the weights $a_i$ into the vector $w$ We will consider networks that are fully connected at each level $l$. I.e. we will consider the set of mappings between layers:

$$x \to x^{(1)} \to x^{(2)} \ldots \to x^{(k)} \tag{3}$$

where the $j$:th output node at the $l + 1$:th layer is computed as:

$$x_j^{(l+1)} \;=\; [w_j^{l+1^T} x^{(l)}]_+ \tag{4}$$

This is a somewhat generalised model compared to the more standard convolutional networks with multiple kernels at each level. It can however be easily specialised to the convolutional case which we will will do later.

For each point $x^{l+1}$ in the activity output space of level $l+1$ we can define the *preimage set* $P(x^{l+1})$ of activities $x^l$ at level $l$ that maps to this activity. We can illustrate this as in Figure 1 with examples of preimages of the mapping between successive layers for a simple 2-node network. If there was no non linear rectifying element, the mapping would be just the linear plus bias transformation from layer $l$ to layer $l + 1$ and and it could be read out from the figure by just noting the respective coordinates in the orthogonal $l$-system and the skewed $l + 1$-system.

$$x_1^{(l+1)} \;=\; [w_{1,1}x_1^{(l)} + w_{1,2}x_2^{(l)} + w_{1,3}]_+ \tag{5}$$

$$x_2^{(l+1)} \;=\; [w_{2,1}x_1^{(l)} + w_{2,2}x_2^{(l)} + w_{2,3}]_+ \tag{6}$$

For the all-positive quadrant in the output $(x_1^{(l+1)}, x_2^{(l+1)})$ the mapping is unique and the preimage is just the corresponding input $(x_1^{(l)}, x_2^{(l)})$. The dashes lines in the two other quadrants illustrate the preimage for the case of points on the lines $x_1^{(l+1)} = 0$ and $x_2^{(l+1)} = 0$ respectively. The grey area in the last quadrant indicate points that all get mapped to the origin $(0, 0)$ of the $(x_1^{(l+1)}, x_2^{(l+1)})$ coordinate system.

$$\begin{pmatrix} x_1^{l+1} \\ x_2^{l+1} \\ 1 \end{pmatrix} = \begin{pmatrix} w_{1,1}^{(l+1)} & w_{1,2}^{(l+1)} & w_{1,3}^{(l+1)} \\ w_{2,1}^{(l+1)} & w_{2,2}^{(l+1)} & w_{2,3}^{(l+1)} \\ 0 & 0 & 1 \end{pmatrix} \begin{pmatrix} x_1^l \\ x_2^l \\ 1 \end{pmatrix}$$

By concatenating this these transformations from the first layer we can express the activity at level $(l + 1)$ directly as am affine mapping from the input space $(x1, x2)$

$$\begin{pmatrix} x_1^{l+1} \\ x_2^{l+1} \\ 1 \end{pmatrix} = \begin{pmatrix} w_{1,1}^{(l+1)} & w_{1,2}^{(l+1)} & w_{1,3}^{(l+1)} \\ w_{2,1}^{(l+1)} & w_{2,2}^{(l+1)} & w_{2,3}^{(l+1)} \\ 0 & 0 & 1 \end{pmatrix} \begin{pmatrix} w_{1,1}^{(l)} & w_{1,2}^{(l)} & w_{1,3}^{(l)} \\ w_{2,1}^{(l)} & w_{2,2}^{(l)} & w_{2,3}^{(l)} \\ 0 & 0 & 1 \end{pmatrix} \cdots \begin{pmatrix} w_{1,1}^{(1)} & w_{1,2}^{(1)} & w_{1,3}^{(1)} \\ w_{2,1}^{(1)} & w_{2,2}^{(1)} & w_{2,3}^{(1)} \\ 0 & 0 & 1 \end{pmatrix} \begin{pmatrix} x_1 \\ x_2 \\ 1 \end{pmatrix}$$

In fig 2 the coordinate systems depicts these mappings from the input space $(x_1, x_2)$ to the node activities and levels 1, 2 and 3 and how the preimages at the output level can be traced as elements in input space forming piecewise linear manifolds Note that disjoint piecewise linear regions in input space $(x_1, x_2)$ gets mapped to distinct points on the two output axis $(x_1^{(3)}, x_2^{(3)})$ while all the points in the grey shaded area get mapped to the same point $x_1^{(3)} = 0, x_2^{(3)} = 0$ This illustrates the network's ability to map non-linear input regions into linear output regions which is essential for successful classification.

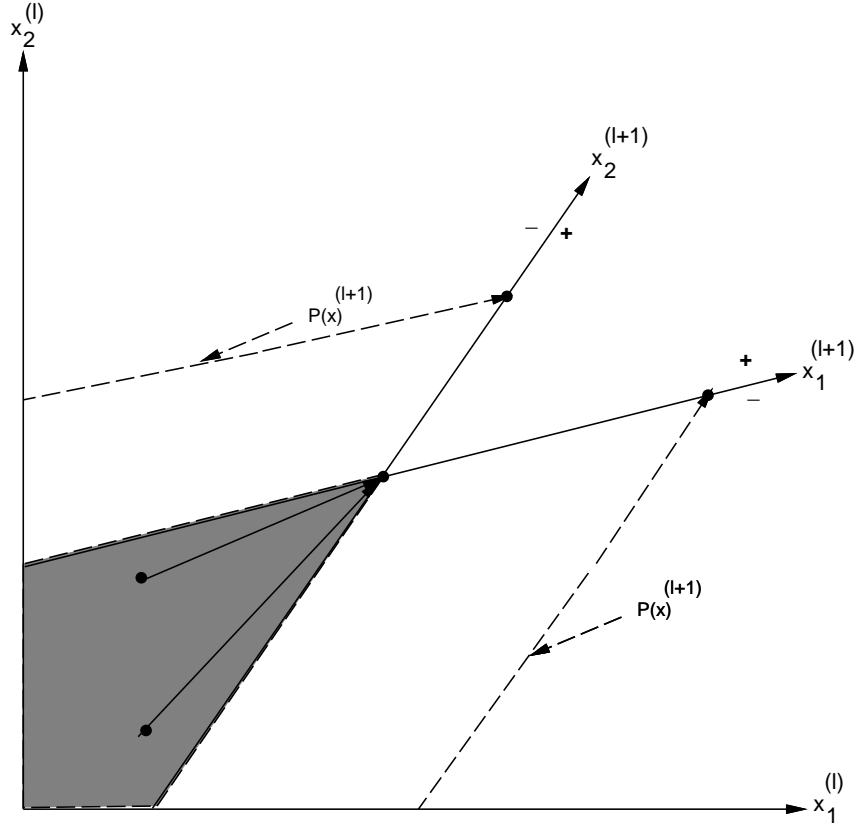

Figure 1: Mapping from activity $x$ at level $l$ to level $l+1$ and the associated preimage sets $P(x)$. The figure illustrates the two coordinate axis' $x_1^{(l+1)} = 0$ and $x_2^{(l+1)} = 0$. In the all positive quadrant, the mapping is just linear while in the other quadrants, the input gets mapped to 0 output of either $x_1^{(l+1)}$ or $x_2^{(l+1)}$ or both. This generates preimage sets depending on the quadrant.

## 2  GENERAL FULLY CONNECTED RECTIFIER NETWORKS

We collect the linear mappings at level $l$ in the matrix $W$. The number of rows of this matrix is the dimensionality of the output which we can be varying but we will assume that $W$ always has full rank in order to focus on problems induced by the non linear ReLU element. If we denote by $[x]_+$ the output vector with component wise application of the ReLU function on the vector $x$, we then can write:

$$x^{(l+1)} = [Wx^{(l)}]_+$$

for the mapping of activities from layer $l$ to layer $l+1$. For each element $x^{(l+1)}$ the preimage set of this mapping will be the set:

$$P(x^{(l+1)}) = \{x : x^{l+1} = [Wx]_+\}$$

which can be specified in more detail as:

$$P(x^{(l+1)}) = \{x : w_i^T x = x_i^{l+1} \ \forall x_i^{l+1} > 0, \ w_i^T x \le 0 \ \forall x_i^{l+1} = 0\}$$

Let $i_1, i_2, \ldots i_p$ be the indices of the components of $x^{l+1}$ that are $= 0$ and $j_1, j_2, \ldots j_q$ those that are $> 0$. This means that

$$w_{i_1}^T x^{(l)} \le 0, \quad w_{i_2}^T x^{(l)} \le 0 \quad \ldots \quad w_{i_p}^T x^{(l)} \le 0 \tag{7}$$

$$w_{j_1}^T x^{(l)} > 0, \quad w_{j_2}^T x^{(l)} > 0 \quad \ldots \quad w_{j_p}^T x^{(l)} > 0$$

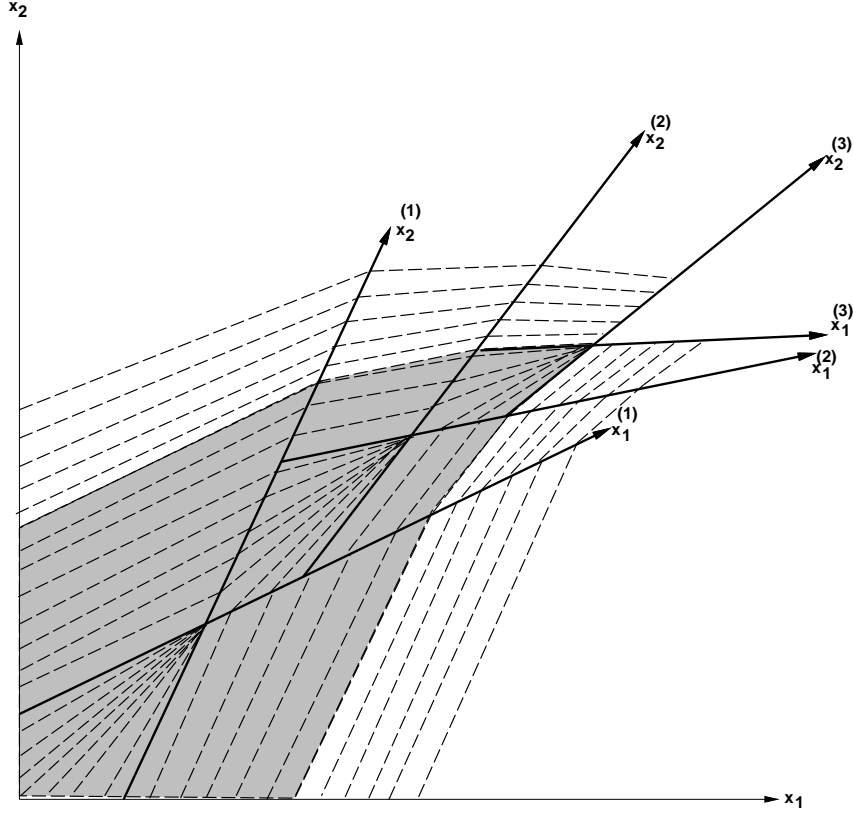

Figure 2: Preimages at various levels of a rectifier network with input $(x_1, x_2)$ and output activity $(x_1^{(3)}, x_2^{(3)})$ All elements in the grey shaded area eventually get mapped to output activity $(0, 0)$ and are irreversibly mixed.

For the case $p = 0$ we have a trivial linear mapping from the previous layer to only positive values of the output. This means that the preimage is just the point $x^{(l)}$. In the general case where $p > 0$ the preimage will contain elements $x$ such that $w_i^T x < 0$ for $i_1, i_2, \ldots i_p$. In order to identify these we will define the null spaces of the linear mappings $w_i$:

$$\Pi_i = \{x : w_j^T x = 0 \quad j = 1 \ldots n\}$$

These null spaces are sets of hyperplanes in input space. Obviously, any input element $x$ that is mapped to the negative side of the hyperplane generated by the mapping $w^i$ will get mapped to this hyperplane by the ReLU function. In order to identify this mapping we will define a set of basis vectors for elements of the input space from the one dimensional linear subspaces generated by the intersections:

$$\pi_i = \Pi_1 \cap \Pi_2 \cap \ldots \cap \Pi_{i-1} \cap \Pi_{i+1} \cap \ldots \cap \Pi_n$$

Each one dimensional subspace $\pi_i$ is generated by intersecting the hyperplanes associated with the nullspaces of the remaining linear mapping kernels. The fact that these intersections generate one dimensional subspaces can be seen most easily using e.g. Grassmann-Cayley algebra Carlsson (1993) or by just noting that each intersection of two $n$-dimensional hyperplanes gives rise to a linear manifold with dimension one lower at each intersection For each subspace $\pi_i$ we can now define a basis unit vector $e_i$ such that each element of $\pi_i$ can be expressed as $x = \alpha_i e_i$. We can also define the direction and length of $e_i$ by requiring that $w_i^T e_i = 1$ The assumed full rank of the mapping $W$ guarantees that the system $e_1, e_2 \ldots e_n$ is complete in the input space. We can therefore express any vector as:

$$x = \sum_1^n \alpha_i e_i$$

Since $e_i$ is in the nullspace of every remaining kernel except $i$ we have:

$$w_j^T e_i = 0 \quad i \neq j$$

This means that:

$$w_j^T x = \sum_1^n \alpha_i w_j^T e_i = \alpha_j$$

The subspace coordinates $\alpha_i$ are therefore a convenient tool for identifying the preimage of the mapping between the successive layers in a rectifier network. Since for $j = i_1, i_2, \ldots i_p$ we will have $\alpha_j > 0$ and for $j = j_1, j_2, \ldots j_q$ we will have $\alpha_j \leq 0$

We can therefore finally formulate the procedure for identifying the preimage of a mapping between successive layers in a rectifying network as:

Given the mapping where the activity of the $j$:th node is computed as:

$$x_j^{(l+1)} = [w_j^T x^{(l)}]_+ \tag{8}$$

we identify indices $j = i_1, i_2, \ldots i_p$ where $w_j^T x^{(l)} > 0$ and $j = j_1, j_2, \ldots j_q$ where $w_j^T x^{(l)} \leq 0$ Using kernels $w_1 \ldots w_n$ to define their corresponding null-space hyperplanes $\Pi_1 \ldots \Pi_n$ we generate one dimensional subspaces $\pi_i$ by intersecting the complementary set of null-space hyperplanes:

$$\pi_i = \Pi_1 \cap \Pi_2 \cap \ldots \cap \Pi_{i-1} \cap \Pi_{i+1} \cap \ldots \cap \Pi_n$$

and define basis vectors for these as $e_i$ Any element in the input space can now be expressed as a linear combination:

$$x = \alpha_{i_1} e_{i_1} + \alpha_{i_2} e_{i_2} + \ldots \alpha_{i_p} e_{i_p} - \alpha_{j_1} e_{j_1} - \alpha_{j_2} e_{j_2} - \ldots \alpha_{j_q} e_{j_q}$$

where all $\alpha_i \geq 0$ The preimage set is then generated by assigning arbitrary values $> 0$ to the coefficients $\alpha_{j_1}, \alpha_{j_2}, \ldots \alpha_{j_q}$

Figure 3 illustrates the associated hyperplanes $\Pi_1, \Pi_2, \Pi_3$ in the case of three nodes and the respective unit vectors $e_1, e_2, e_3$ with positive directions indicated by arrows. For the all positive octant, i.e. all $w_i^T x > 0$ the linear mapping is just full rank and the preimage is just the associated input $(x_1, x_2, x_3)$. For three other octants the preimages for three selected points are illustrated:

1. For $w_1^T x > 0, w_2^T x > 0, w_3^T x < 0$, the preimage of a point on the plane $\Pi_3$ consist of all points on the indicated arrow.

2. For $w_1^T x > 0, w_2^T x < 0, w_3^T x > 0$, the preimage of a point on the plane $\Pi_2$ consist of all points on the indicated arrow.

3. For $w_1^T x > 0, w_2^T x < 0, w_3^T x < 0$, the preimage of a point on the intersection of planes $\Pi_2$ and $\Pi_3$ consist of all points on the indicated grey shaded area

In general, for points that are not in the all positive $w_i^T x > 0 \forall i$ region they will be located on a linear submanifold spanned by the unit vectors $e_{i_1}, e_{i_2}, \ldots, e_{i_p}$

$$x = \alpha_{i_1} e_{i_1} + \alpha_{i_2} e_{i_2} + \ldots \alpha_{i_p} e_{i_p}$$

The preimage then consists of all points on the linear manifold:

$$x - \alpha_{j_1} e_{j_1} - \alpha_{j_2} e_{j_2} - \ldots \alpha_{j_q} e_{j_q}$$

where all $\alpha_i \geq 0$

For a multi level network , preimages for elements that are mappings between successive levels will therefore consist of pieces of linear manifolds in the input space at that level of dimensions determined by the number of nodes with positive output for that element. By mapping back to the original input space, preimages for specific elements at a certain level will be piecewise linear manifolds, the elements of which all map to that specific element. This is exactly what is illustrated in figure 2 for the case of 2-dimensional inputs and a network with three levels of two nodes at each level.These piecewise linear manifolds can therefore me considered as fundamental building blocks for mapping input distributions to node outputs at any level of the network.

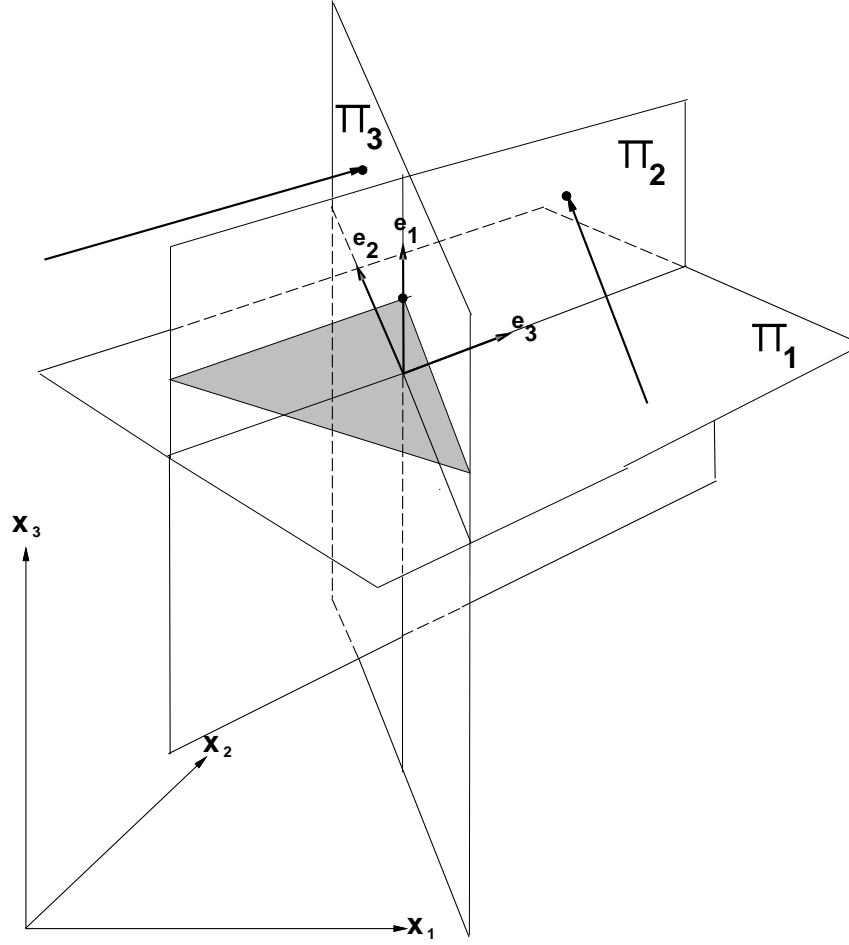

Figure 3: Hyperplanes $\Pi_1, \Pi_2 \Pi_3$ of nullspaces for transformation kernels and the associated unit vectors $e_1, e_2, e_3$ from pairwise intersections $(\Pi_2 \Pi_3)$ $(\Pi_1, \Pi_3)$ and $(\Pi_1, \Pi_2)$ respectively. The preimages of various points in the output are indicated as arrows or the shaded area

## 3 CONVOLUTIONAL NETWORKS HAVE "WELL-BEHAVED" PREIMAGES

Convolutional networks where the mappings $W$ consists of convolutional matrices:

$$
\begin{pmatrix}
w^T & 0 & \ldots & & & & 0 \\
0 & w^T & 0 & \ldots & & & 0 \\
0 & 0 & w^T & 0 & \ldots & & 0 \\
0 & 0 & 0 & w^T & 0 & \ldots & 0 \\
0 & & & & \vdots &
\end{pmatrix}
\tag{9}
$$

are the standard realisations of multilayer networks. Using a heuristic argument, it can be seen that the preimages of these networks are in general "well-behaved" in the sense that under very general assumptions that are typically valid when training these networks the preimages generated for a specific activity associated with an image will be semantically equivalent to the given image. For any layer, the accumulated kernel $w^{(l)}$ mapping from the input space will be the concatenation with the specific kernel $w'^{(1)}$ of that layer and all specific kernels from the previous layers. I.e. they are generated by repeated convolutions:

$$
w^{(l)} = w'^{(1)} * w'^{(2)} * \ldots * w'^{(l-1)} * w'^{(l)}
\tag{10}
$$

of the specific kernels $w'^{(i)}$ from the previous layers. In general the kernels associated with lower levels of the network will be associated with features such as edges in various orientation. They

are therefore in general not responsive to slowly varying signal intensity inputs. Typically they will therefore have the constant vector $(1, 1, \ldots 1)$ as nullvector. Any nullvector of a kernel at a certain level will also be a nullvector of the kernels associated with higher levels. If we study the nullvector $(x_1 \ldots x_n)$ associated with the complementary set of kernels when deleting a specific kernel $(i)$ so that no overlap occurs between kernels $(i - 1)$ and $(i + 1)$

$$
\begin{pmatrix}
w^T & 0 & \ldots & & & & 0 \\
0 & w^T & 0 & \ldots & & & 0 \\
0 & 0 & w^T & 0 & \ldots & & 0 \\
0 & 0 & 0 & w^T & 0 & \ldots & 0 \\
0 & & & & \vdots & &
\end{pmatrix}
\begin{pmatrix}
x_1 \\ x_2 \\ \vdots \\ x_n
\end{pmatrix}
= 0
\tag{11}
$$

it can easily be seen that

$$
(x_1, x_2 \ldots x_n) = (a, \ldots a, b \ldots, b)
\tag{12}
$$

i.e. a " step edge " where the step occurs at the location $(i)$ of the deleted kernels. Sharp step edges in an image will lead to high likelihood of the convolution being negative. The preimages associated with an image with step edges therefore consists of the original image with step edges overlaid on already existing edges. This will in general not change the semantic content of the image. At higher levels, negative convolutions and nullspaces will be associated with more complex image structures and we can expect more complex variations in the preimage set such as the blurrings associated with the results in Mahendran & Vedaldi (2015; 2016) This suggests that the preimages associated with images in standard convolutional networks are not adversarial but rather fall in the same class as that of the image in question. It will be a focus of further work to outline in more detail the structure of the preimage class of a certain input image in order to find out if it represents e.g variations due to external factors and/or if it contains truly adversarial input exemplars. An interesting possibility would be that the set of preimages can be considered as a model set for any class of images and that the ultimate goal of training a network will be to have the set of preimages of the output nodes of a certain class coincide with the image manifold associated with that class.

## 4 IMPLICATIONS FOR MODELLING IMAGE MANIFOLDS

### 4.1 THE "GENERAL IMAGE MANIFOLD"

It is generally believed that a valid model for the distribution of images of various identifiable classes is that they lie on relatively low dimensional manifolds in the high dimensional image input space. One can also think about the "general image manifold" as consisting of all possible images that contain identifiable visual structures. The manifolds associated with specific object classes would then be contained in this general manifold. Due to external factors associated with the imaging situation like viewpoint, illumination , shading etc. the elements of a specific class will be distributed in input image space on the specific object manifold.

### 4.2 CONSTRAINTS ON KERNELS FOR AVOIDING THE MIXING OF PREIMAGES

Given a specific assumption of the distribution of classes on the image manifold we can postulate various properties of the kernels of rectifier networks at various levels that are necessary in order that the preimages associated with elements of different classes should not overlap. If we make the hypothesis that external factors influence different classes in a similar way we can expect the variation of the class manifolds due to external factors to be similar. Fig 4 depicts a simplified hypothetical case of the input image manifold where the variation due to external factors has been reduced to just one-dimension.

Figure 5 illustrates the placement of the hyperplanes associated with kernels at various levels in a network that would be necessary in order for the preimages of the classes not to mix with each other. Due to the assumed similarity of the variation of the individual class manifolds, the hyperplanes have to be orthogonal to the external factor variation. This highly simplified sketch just illustrates the basic idea that the covariation of class manifolds would induce severe constraints on the kernels at various levels in a network

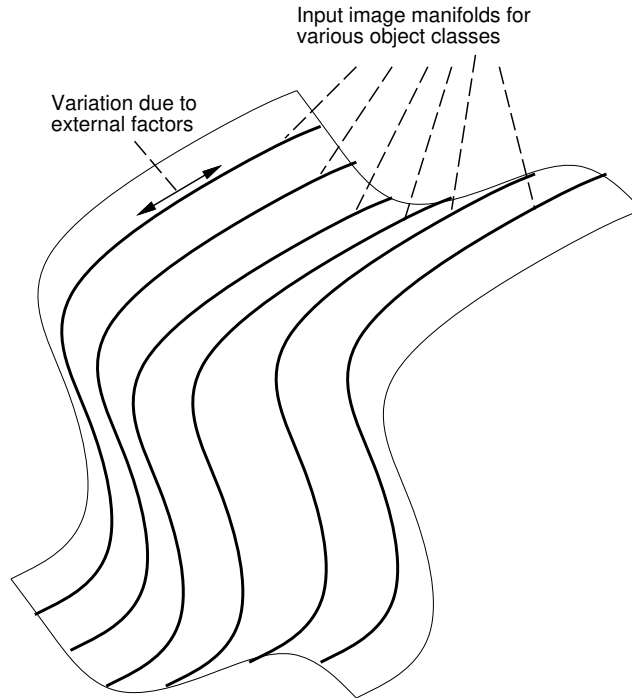

Figure 4: The "general image manifold" and manifolds of individual object classes due to external factor and intra class variation assuming a high degree of covariance between classes due to external factors

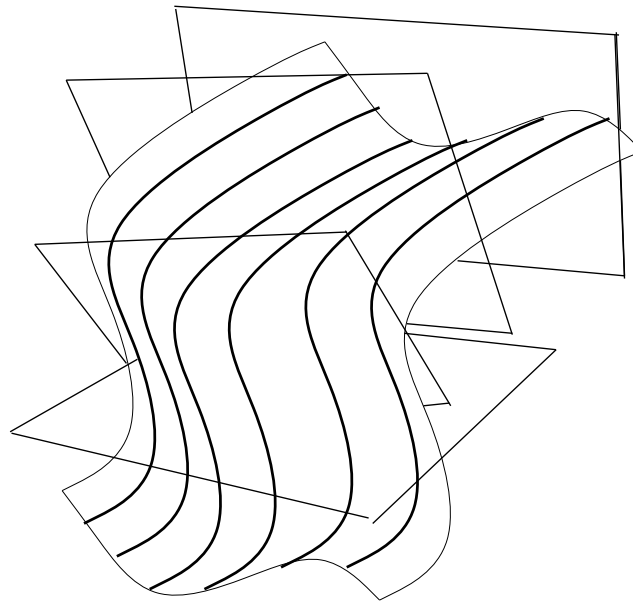

Figure 5: Hyperplanes associated with kernels at various levels in a rectifier network that will avoid mixing of preimages from different classes.

The covariation of different object classes would have the important implication that the training of the kernels at these levels of the network would benefit from exemplars of multiple classes. This would give an explanation for the relative success of deep learning methods compared to previous approaches that in general relied on training of individual classes.

### 4.3 SEPARATING CLASSES

At the output level of a network, the representation of of various classes have to be disentangled in order for the final fc layer to perform a linear separation of the classes. We have empirically found that classes at the final layer of a network are highly concentrated on a small set sometimes individual nodes. The whole network can be seen as a dimensionality reducing device that starts out in high dimensional image input space and ends up in very low dimensional output layers. This dimensionality reduction is achieved by the ReLU units at various layers as illustrated in fig. 3 where the preimage, being a linear manifold is mapped from a higher to a lower dimension. At various levels of the network, different classes can located on the same manifold or the same class can be distributed on different manifolds. This means that we need procedures for ReLU networks that splits different classes on the same manifold or merges a class that is represented on several manifolds to one and the same. This can be achieved by ReLU networks and fig. 6 illustrates for the simple 2 node network how the preimages of classes can be split or merged by proper selection of kernels at successive levels.

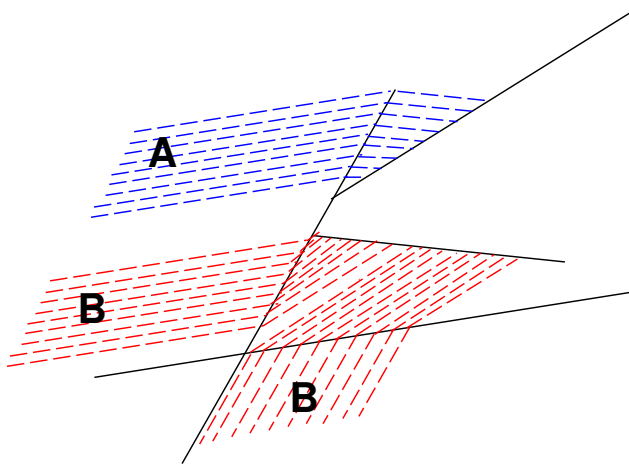

Figure 6: Illustration how different classes A and B on the same manifold can be split to different manifolds and how the same class B on different manifolds can be merged to the same manifold

In summary we have discussed how the properties of a network regarding it's ability to model input manifolds and achieve efficient classification can be described with the concept of the preimage of the activities of the rectifier network at a certain level. For a specific class the set of preimages expressed in input image space resulting from the totality of activities at the output of the network constitutes the network's model of the manifold of that class. The efficient training of a network can be seen as obtaining the correct model of every class to be discriminated. The preimage concept allows us to describe how the network is constrained by properties of these manifolds and the requirement of obtaining efficient classification at the end.

## 5 CONCLUSIONS AND FURTHER WORK

We have described a procedure to compute the preimage of the activity at a certain level in a deep network. I.e the set of inputs to the level that result in the same output activity. By concatenating this procedure we can compute the preimage at input level, i.e. the set of input exemplars that will eventually result in the same activity. Since inputs in the same preimage of any level activity are irreversibly mixed, they should ideally correspond to exemplars in classes to be discriminated. They therefore constitute building blocks for capturing the manifolds of classes in input space. The fact that deep networks can be seen as tools for efficient low dimensional piecewise linear approximation has been pointed out in other works recently Basri & Jacobs (2016) and is an important component in understanding how deep networks achieves their unprecedented efficiency Brahma et al. (2016); An et al. (2015) .

It will also be the objective of further work to investigate empirically if the assumed models of image manifolds and their relation to preimages are valid. This will involve the actual computation of preimage manifolds which essentially involves the computatio of nullspaces of weight matrices at various levels in order to define basis vectors for the manifolds. A deeper analysis of the preimage problem will also have to deal with the pooling that takes place in a deep learning network.

It will also be interesting to investigate how knowledge of preimages in deep networks can be used to enhance the efficiency of the training of the network. In order to do this we will have to consider the specialisation to convolutional layers and what it implies. It will also be relevant to investigate the possible nature of adversarial exemplars Szegedy et al. (2013); Goodfellow et al. (2014); Nguyen et al. (2015) in classification and if they are related to the concept of pre image of activities associated with specific classes.

ACKNOWLEDGMENTS

We would like to thank Andrew Zisserman for pointing out the work of Mahendran & Vedaldi (2015; 2016). Stefan Carlsson, Hossein Azizpour and Ali Razavian are supported by the School of Computer Science and Communication, KTH. In Razavian's case via a grant held by Atsuto Maki

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
