# Peer review of "The Preimage of Rectifier Network Activities"

_ICLR 2017 — rejected_

[Reviewer Comment · AnonReviewer3 · rating 4 · 18 Dec 2016]
**not ready yet**

I really appreciate the directions the authors are taken and I think something quite interesting can come out of it. I hope the authors continue on this path and are able to come up with something quite interesting soon. However I feel this paper right now is not quite ready. Is not clear to me what the results of this work are yet. The preimage construction is not obviously (at least not to me) helpful. It feels like the right direction, but it didn't got to a point where we can use it to identify the underlying mechanism behind our models. We know relu models need to split apart and unite different region of the space, and I think we can agree that we can construct such mechanism (it comes from the fact that relu models are universal approximators) .. though this doesn't speak to what happens in practice.  All in all I think this work needs a bit more work yet.

[Official Review · AnonReviewer1 · rating 4 · confidence 5 · 19 Dec 2016]
**review of ``THE PREIMAGE OF RECTIFIER NETWORK ACTIVITIES''**

I have not read the revised version in detail yet. 

SUMMARY 
This paper studies the preimages of outputs of a feedforward neural network with ReLUs. 

PROS 
The paper presents a neat idea for changes of coordinates at the individual layers. 

CONS 
Quite unpolished / not enough contributions for a finished paper. 

COMMENTS 
- In the first version the paper contains many typos and appears to be still quite unpolished. 

- The paper contains nice ideas but in my opinion it does not contribute sufficiently many results for a Conference paper. 
I would be happy to recommend for the Workshop track. 

- Irreversibly mixed and several other notions from the present paper are closely related to the concepts discussed in [Montufar, Pascanu, Cho, Bengio, NIPS 2014]. I feel that that paper should be cited here and the connections should be discussed. In particular, that paper also contains a discussion on the local linear maps of ReLU networks. 

- I am curious about the practical considerations when computing the pre-images. The definition should be rather straight forward really, but the implementation / computation could be troublesome. 


DETAILED COMMENTS 
- On page 1 ``can easily be shown to be many to one'' in general. 

- On page 2 ``For each point x^{l+1}'' The parentheses in the superscript are missing. 

- After eq. 6 ``the mapping is unique''  is missing `when w1 and w2 are linearly independent' 

- Eq. 1 should be a vector. 

- Above eq. 3. ``collected the weights a_i into the vector w'' and bias b. Period is missing. 

- On page 2 ``... illustrate the preimage for the case of points on the lines ... respectively'' 
Please indicate which is which.  

- In Figure 1. Is this a sketch, or the actual illustration of a network. In the latter case, please state the specific value of x and the weights that are depicted. Also define and explain the arrows precisely. 
What are the arrows in the gray part? 

- On page 3 `` This means that the preimage is just the point x^{(l)}''  the points that W maps to x^{(l+1)}. 

- On page 3 the first display equation. There is an index i on the left but not on the right hand side. 
The quantifier in the right hand side is not clear. 

- ``generated by the mapping ... w^i '' subscript

- ``get mapped to this hyperplane'' to zero 

- ``remaining'' remaining from what? 

- ``using e.g. Grassmann-Cayley algebra'' 
How about using elementary linear algebra?!

- ``gives rise to a linear manifold with dimension one lower at each intersection'' 
This holds if the hyperplanes are in general position. 

- ``is complete in the input space'' forms a basis 

- ``remaining kernel'' remaining from what? 

- ``kernel'' Here kernel is referring to nullspace or to a matrix of orthonormal basis vectors of the nullspace, or to what specifically? 

- Figure 3. Nullspaces of linear maps should pass through the origin. 

- `` from pairwise intersections'' \cap 

- ``indicated as arrows or the shaded area'' this description is far from clear. 

- typos: peieces, diminsions, netork, me,

[Official Review · AnonReviewer2 · rating 4 · confidence 4 · 20 Dec 2016]
**Nice premise but somewhat incomplete?**

Summary:
This paper looks at the structure of the preimage of a particular activity at a hidden layer of a network. It proves that any particular activity has a preimage of a piecewise linear set of subspaces.

Pros:
Formalizing the geometry of the preimages of a particular activity vector would increase our understanding of networks

Cons:
Analysis seems quite preliminary, and no novel theoretical results or clear practical conclusions.

The main theoretical conclusion seems to be the preimage being this stitch of lower dimensional subspaces? Would a direct inductive approach have worked? (e.g. working backwards from the penultimate layer say?) This is definitely an interesting direction, and it would be great to see more results on it (e.g. how does the depth/width, etc affect the division of space, or what happens during training) but it doesn't seem ready yet.

[Final Decision · Program Chairs · 06 Feb 2017]
**ICLR committee final decision**

This paper studies the invertibility properties of deep rectified networks, and more generally the piecewise linear structure that they implicitly define. The authors introduce a 'pseudocode' to compute preimages of (generally non-invertible) half-rectified layers, and discuss potential implications of their method with manifold-type models for signal classes. 
 
 The reviewers agreed that, while this is an interesting and important question, the paper is currently poorly organized, and leaves the reader a bit disoriented, since the analysis is incomplete. The AC thus recommends rejection of the manuscript. 
 
 As an addendum, the AC thinks that the authors should make an effort to reorganize the paper and clearly state the contributions, and not expect the reader to find them out on their own. In this field of machine learning, I see contributions as being either (i) theoretical, in which case we expect to see theorems and proofs, (ii) algorithmical, in which case an algorithmic is presented, studied, extensively tested, and laid out in such a way that the interested reader can use it, or (iii) experimental, in which case we expect to see improved numerical performance in some dataset. Currently the paper has none of these contributions. 
 Also, a very related reference seems to be missing: "Signal Recovery from Pooling Representations", Bruna ,Szlam and Lecun, ICML'14, in which the invertibility of ReLU layers is precisely characterized (proposition 2.2).